# The Diverse Biological Activity of Recently Synthesized Nitro Compounds

**DOI:** 10.3390/ph15060717

**Published:** 2022-06-05

**Authors:** Saúl Noriega, Jaime Cardoso-Ortiz, Argelia López-Luna, Ma Del Refugio Cuevas-Flores, Juan Armando Flores De La Torre

**Affiliations:** Unidad Académica de Ciencias Químicas, Universidad Autónoma de Zacatecas, Zacatecas 98160, Mexico; saul.noriega@uaz.edu.mx (S.N.); mariaa.lopez@uaz.edu.mx (A.L.-L.); qkis.cuevas@uaz.edu.mx (M.D.R.C.-F.); armando.flores@uaz.edu.mx (J.A.F.D.L.T.)

**Keywords:** antimicrobial, vasodilatory, anti-inflammatory, nitro group

## Abstract

The search for new and efficient pharmaceuticals is a constant struggle for medicinal chemists. New substances are needed in order to treat different pathologies affecting the health of humans and animals, and these new compounds should be safe, effective and have the fewest side effects possible. Some functional groups are known for having biological activity; in this matter, the nitro group (NO_2_) is an efficient scaffold when synthesizing new bioactive molecules. Nitro compounds display a wide spectrum of activities that include antineoplastic, antibiotic, antihypertensive, antiparasitic, tranquilizers and even herbicides, among many others. Most nitro molecules exhibit antimicrobial activity, and several of the compounds mentioned in this review may be further studied as lead compounds for the treatment of *H. pylori, P. aeruginosa, M. tuberculosis* and *S. mutans* infections, among others. The NO_2_ moiety triggers redox reactions within cells causing toxicity and the posterior death of microorganisms, not only bacteria but also multicellular organisms such as parasites. The same effect may be present in humans as well, so the nitro groups can be considered both a pharmacophore and a toxicophore at the same time. The role of the nitro group itself also has a deep effect on the polarity and electronic properties of the resulting molecules, and hence favors interactions with some amino acids in proteins. For these reasons, it is fundamental to analyze the recently synthesized nitro molecules that show any potential activity in order to develop new pharmacological treatments that enhance human health.

## 1. Introduction

Pharmaceuticals are chemical substances able to cause a local or systemic action in humans and animals. A specific chemical structure is fundamental regarding its activity since the molecule relates to specific targets due to several interactions between the pharmaceutical itself and receptors, enzymes or channels. After absorption, pharmaceuticals may undergo some structural changes with the consecutive change in activity, which can increase, decrease or even change. Since the discovery of chloramphenicol in 1947, nitro containing cycles and heterocycles have been extensively investigated due to their antibacterial activity; however, nitro containing molecules exhibit a wide variety of biological activities that are attractive to medicinal chemists, for example, some are antineoplastic, antibiotic, antihypertensive, antiparasitic agents, tranquilizers and even herbicides, among many others [1,2,3,4,5,6,7]. The nitro group (NO_2_) is a functional group formed by one nitrogen atom linked to two oxygens, it is a particularly electron-withdrawing moiety since the N has no lone pair, hence it bears a positive charge (Figure 1) [8]. The electron-withdrawing effect is easily observed in aromatic rings **1** due to resonance with the nitro group, deactivating certain positions and causing changes in the polarity of molecules, which in some cases favors interaction with nucleophilic sites of protein structures such as enzymes, causing inhibition. The nitro group not only works as a pharmacophore but also as a toxicophore, making it even more interesting for medicinal chemists. In most cases, the nitro group is present in a specific structure but it is not entirely related to the activity; however, it affects pharmacokinetics. The whole activity regarding the nitro group directly (whether beneficial or toxic) depends entirely on the reduction of the nitro itself, accepting up to six electrons in order to form the amine derivative. This reduction occurs through enzymatic reactions using NADH or NADPH as reducing agents. During the reduction of the NO_2_, some nitroso and hydroxylamine intermediates are formed and react with biomolecules producing undesired toxic and mutagenic effects. Although reduction of the nitro group itself is toxic to humans, in some cases this might be seen as the desired effect for designing and synthesizing new antiparasitic drugs. In some other cases (most of them) the activity of the corresponding molecule is “affected” just by the presence of the NO_2_ and its electron-withdrawing properties, polarity, or stereochemistry. The aim of the present review is to describe representative nitro compounds that show some biological activity, either by having a NO_2_ as a pharmacophore or as a moiety in the molecule indirectly enhancing its activity. This review is focused on research that shows advances in the synthesis and biological activity of nitro-containing drugs; several reports have been analyzed in the period between 2018 and 2022 [9,10,11,12,13,14,15,16,17,18,19]. 

## 2. Antimicrobial Activity

Nitro-containing molecules are some of the first lines of treatment for common infections caused by several microorganisms. Metronidazole, chloramphenicol and other nitro derivatives display antimicrobial activity by different mechanisms. One of the most accepted and general models states that when nitro compounds are reduced, they produce toxic intermediates (such as nitroso and superoxide species), then, the reduced nitro species bind covalently to DNA resulting in nuclear damage and cell death [20]. Some examples of molecules that required activation in order to be antimicrobial agents are 5-nitroimidazole derivatives. These compounds undergo intercellular reduction giving rise to a nitro anion radical (NO_2_^−^) of short life but a crucial step in the mechanism of action of nitroimidazoles. 5-nitroimidazole is a significant moiety in medicinal chemistry since it is present in commercial drugs such as miconazole, ketoconazole and metronidazole among others [13,21]. 

### 2.1. Antibacterial

The antibacterial activity of nitro-containing molecules is one of the widest effects observed, not only in human or veterinary pharmaceuticals but also in the manufacturing of new materials with antimicrobial properties [22]. Metronidazole is one of the primary pharmaceuticals employed in the treatment of *H. pylori*, and it contains the moiety 5-nitroimidazole. Unfortunately, it has lost efficiency as this microorganism is developing resistance against traditional treatments. For this reason, several 5-nitroimidazole derivatives have been developed throughout the years in order to overcome this problem, the imidazole base structure is known for having activity against *H. pylori* as it can be found already existing pharmaceuticals such as metronidazole (Figure 2) [23]. 

Many existing drugs have some degree of nitration in their structures, for example, chloramphenicol or nitrofurantoin. These well-known substances are of common use and will continue so for a while; however, there are many other compounds being developed and tested for their antimicrobial properties. Such is the case of a series of nitrated pyrrolomycins that proved to be efficient against Gram-negative (*Pseudomonas aeruginosa*) and Gram-positive (*Staphylococcus aureus*) bacteria [24]. Pyrrolomycins are halogenated organic compounds isolated from *Actinosporangium* and *Streptomyces* species that show a variety of biological activities such as neuro and immune-modulatory, antiproliferative, insecticidal and antimicrobial. This latter property is attributed to a protonophoric effect of pyrrolomycins, which is the ability of some molecules to translocate protons from one side to another of biological membranes. Even though pyrrolomycins are potential candidates for developing new antibacterial pharmaceuticals, they are also toxic compounds due to the halogens present in their structure (Cl or Br) [25]. By chemically modifying the structure of some pyrrolomycins, researchers are able to obtain new compounds with enhanced antimicrobial activity and low toxicity. One of these modifications includes a nitration step on the pyrrole ring **3a–d**, which is easily achieved through an electrophilic aromatic substitution reaction (Figure 3). The C2 position in the pyrrole ring is favored for electrophilic substitution, however, when C2 has either Br or Cl, C1 becomes particularly activated for an electrophilic attack due to the halogen lone pair. The bromine-substituted compound **4c** showed the best yield compared to Cl derivatives **4b** and **4d**, indicating a stronger activation by Br. Experimental results also indicated that the presence of nitro groups (especially in the C2 and C4 positions) in the pyrrole ring enhanced the antibacterial activity **4b** (20 μM against *S. aureus*) and **4d** (30 μM against *P. aeruginosa*). Researchers propose not only a protonophoric effect, but also improved lipophilicity and hence a better interaction with membranes [24]. 

For many years, the coordination chemistry of transition metals was not considered an important part of medicinal chemistry. However, in recent years, metal complexes with Schiff bases as ligands have gained attention due to their wide potential biological activity. A series of 4-nitro-1,2-phenylendiamine metal complexes were synthesized and tested against Gram-positive microorganisms (*Streptococcus mutans* and *Staphylococcus aureus*) Gram-negative (*Escherichia coli, Klebsiella pneumonia* and *Pseudomonas aeruginosa*) and the fungus *Candida albicans*. A total of nine new compounds were obtained but only the Zn(II) complex **5** showed significant activity against *Streptococcus mutans* (Figure 4). Antibacterial activity was reported as the inhibition zone diameter, with 40.7 mm being the inhibition obtained with the Zn(II) complex. The rest of the compounds with other metals such as Co(II), Cr(III), Fe(III) and Ni(II) showed little or no activity at all [26]. Even when these results might seem discouraging, these kinds of compounds represent new synthetic methods to be addressed in future research or metal complexes with biological activity.

There is an increasing interest in developing new antimicrobial agents not only because of the growing resistance to antibiotics but also because of the risk of nosocomial infections (infectious diseases acquired within a hospital). In some regions, these types of infections could reach up to 40% of hospitalized patients. Gram-positive bacteria are responsible for more than 90% of nosocomial infections, and *Staphylococcus aureus* is one of the pathogens listed as a priority for research and development according to the World Health Organization (WHO). In this context, researchers synthesized several nitro derivatives through an oxa-Michael-aldol condensation reaction and tested their potential antimicrobial activity against *Staphylococcus aureus* and *Candida* sp., two main microorganisms involved in frequent nosocomial infections (Figure 5). Experimental results indicated that nitro derivatives with some degree of halogenation **9b–9d** had the best results with minimum inhibitory concentration (MIC) in the range of 15.6–62.5 μg/mL for *S. aureus* and minimum fungicidal concentration (MFC) 15–500 62.5 μg/mL in the case of *Candida* [27].

*Pseudomonas aeruginosa* is another type of bacteria responsible for nosocomial infections, so it has become important to develop new antibiotics able to inhibit the growth or kill Gram-negative organisms as well. In this matter, heterocycles such as benzothiazoles show antibacterial activity, especially the ones containing the nitro functional group. In a recent study, several nitrated benzothiazoles were obtained through nitroanilines and KSCN under relatively mild conditions. A total of nine compounds were tested against *P. aeruginosa* but only three showed significant activity **10–12** (Figure 6), becoming pharmaceutical targets for future research as lead compounds. Inhibition was compared with procaine penicillin obtaining similar inhibition [28]. 

### 2.2. Antifungal

Nowadays, fungi resistance is becoming a topic of discussion among medicinal chemists. There are some species resistant to classical treatments such as fluconazole (including *C. krusei*, and *C. albicans*), so new molecules are needed in order to limit this growing clinical problem. One of the best methods to obtain new molecules with a desired and specific activity is from already existing molecules that show this particular activity. In this way, researchers are able to obtain new substances by synthesizing derivatives from a parent compound. By these means, a series of fluconazole derivatives **17a–e** were prepared to have a piperazine or nitrotriazole moiety in their structure (Figure 7) [29]. 

The nitrotriazole derivatives were obtained through a four-step synthesis under relatively mild conditions. Although reflux was employed, there was no need for temperatures higher than 80 °C. Yields varied from a range of 30% to 56%, but considering the products the results were satisfactory. The best antifungal properties were observed among the molecules baring halogens in the benzene ring, especially Cl. Halogens tend to make compounds more lipophilic and at the same time more polar, improving their pharmacokinetics. It is noteworthy that the nitrotriazole derivative showed better activity against *C. krusei* (MIC = 1 μM for 17b) compared to the fluconazole standard. The mechanism of action of these fluconazole derivatives involves the inhibition of the 14α-demethylase, an enzyme necessary for the synthesis of ergosterol in fungi [30]. Such antifungal activity observed in the nitro compounds is attributed to an efficient electrostatic interaction between the NO_2_ and the Fe(II) in the heme group of 14α-demethylase, triggering a strong inhibition of the enzyme. Docking studies were performed in order to complement experimental results [29]. 

### 2.3. Antitubercular

Tuberculosis (TB) is an infectious disease caused by *Mycobacterium tuberculosis*, and it is an issue of growing concern for global health because of concomitant infections with HIV. *M. tuberculosis* has developed resistance to traditional treatments, so it has become imperative to find new substances able to kill or inhibit the growth of this pathogen. Regiments for treatment against TB are long (up to 18 months in the case of multi-resistant bacteria) due to its slow bacterial growth, so it is imperative to develop new drugs targeting different metabolic stages. Some lead compounds have been found such as 5-nitrophenantroline **18** with an MIC value of 0.78 μM against *M. tuberculosis* [31]; in this study, researchers found that the presence of the NO_2_ group in the C5 was essential for antitubercular activity (Figure 8). They replaced it with other withdrawing groups, such as cyano, bromo or chloro, and observed ten times less activity with these substituents; furthermore, they also concluded that the NO_2_ moiety modulates intracellular mechanisms involved in killing bacteria [31].

Some nitro-containing molecules such as 5-nitroimidazooxazine and 5-nitro-imidazooxazole are currently studied in clinical trials in the US [32]. However, there have to be more options available for the treatment of tuberculosis. A promising group of molecules with antitubercular activity is nitrotriazoles; in particular, 3-nitro-1,2,4-triazole-based derivatives **19–23** (Figure 9). It was reported that these classes of compounds inhibit *M. tuberculosis* in the range of 3–50 μM. Compared to other effective and still used antitubercular agents such as isoniazid, the MICs (minimum inhibitory concentration) observed in nitrotriazoles are not so far from this reference; they are 0.3 μM for isoniazid and 3.0 μM for the most active nitrotriazole. These molecules are effective, but they still need to be further investigated in order to obtain new nitrotriazole derivatives that can compete with lead compounds such as isoniazid or rifampicin [33,34]. 

## 3. Anti-Inflammatory Activity

As mentioned before, nitrated molecules show a wide range of biological activities, not only as a pharmacophore (or toxicophoroe) but as a moiety in a particular compound enhancing its pharmacokinetics and, overall, its pharmacodynamic properties. Fatty acids are not only important in the synthesis of membranes and covering the energetic demands of cells, but they also play a fundamental role as precursors of cellular signaling. Nitro fatty acids are molecules naturally present in humans; they are the product of the reaction between electrophilic fatty acids and nitrogen species. These molecules are closely related to inflammation mediators as they can react with specific proteins changing their function during cellular signaling [35,36,37]. Nitrated fatty acids such as 10-NO_2_-OA **24** (oleic acid nitrated in the C10) are known for showing important cytoprotective and anti-inflammatory effects in humans (Figure 10). Even when the structure is relatively simple, the importance of this type of molecule relies on its metabolism. Electrophilic substances such as 10-NO_2_-OA undergo beta oxidation, just as any other fatty acid, giving rise to shorter chains up to eight carbon atoms long. For this reason, it is possible for these electrophilic metabolites to interact in a more polar environment, and as a consequence, a more potent inhibition of the NF-kB factor is achieved [38].

Chronic inflammation is known to be the cause of several metabolic diseases such as obesity, insulin resistance, hypertension, metabolic syndrome, type II diabetes and even cancer [39]. Excessive production of NO during inflammation causes degenerative processes within a cell, so the higher concentration of NO, the greater production of reactive nitrogen oxide intermediates (RNOIs). Under these conditions, NO is synthesized by the inducible nitric oxide synthase (iNOS), so an efficient inhibition of this enzyme is considered a target for developing new anti-inflammatory drugs. Even when the nitro group is considered a “structural alert” due to its elevated toxicity, it can still be part of potent anti-inflammatory agents such as nimesulide [40]. Having this in mind, some nitro substituted benzamides were prepared and tested for their anti-inflammatory activity, particularly for the inhibition of iNOS as well as other pro-inflammatory cytokines such as COX-2, IL1β and TNF-α showing promising results in terms of inhibition (Figure 11) [41].

Nitro benzamides derivatives were obtained under relatively mild conditions using pyridine and/or DMF as solvents and reflux temperature. The mechanism was a nucleophilic substitution reaction and yields were in the range of 55–63%, no purification method was mentioned, and pyridine was employed as both solvent and base under a 10 h reflux. Compound **25f** proved to be not only a potent inhibitor of iNOS, but also a promising multi-target lead compound inhibiting other biomolecules such as COX-2, IL1β and TNF-α. DFT studies suggest that the number of NO_2_, rather than their position in the molecule, favors an inhibitory interaction with the heme fraction of the enzyme. The fact that compound 6 has better anti-inflammatory properties than 5 also suggests that the stereochemistry of the NO_2_ moiety plays an important role in the mechanism of action [41].

## 4. Vasodilatory Activity

Nitro compounds have an outstanding activity associated directly with the NO_2_ moiety, which is the realization of ·NO due to bio-reduction of the NO_2_ fragment in the structure (these compounds are known as “·NO donors”). Nitric oxide causes vasodilation by increasing the concentration of cGMP in vascular smooth muscle cells. An increase of intracellular cGMP also enhances protein kinase G activity resulting in a movement of Ca(II) from the intracellular to extracellular compartments (opening of calcium channels, reuptake of Ca(II) to the sarcoplasmic reticulum). A low concentration of intracellular Ca(II) causes an inhibitory effect on myosin light chains preventing phosphorylation of myosin, hence a vascular smooth muscle cell relaxation resulting in widening of blood vessels. Most of the therapeutics of ·NO donors are focused on the treatment of coronary heart diseases such as angina [42,43,44,45,46]. 

It is known that chronic use of non-steroidal anti-inflammatory drugs (NSAIDs) **26** causes gastrointestinal (GI) damage and an increased risk of cardiovascular events related to vasoconstriction. For this reason, there has been some interest in developing NSAIDs with ·NO donor properties, so they can have the desired analgesic effect and at the same time release ·NO preventing both GI damage and hypertensive crises [47,48,49]. Up to today, ·NO donor NSAIDs have shown a discrete or no effect at all. However, the chemistry behind these nitrocompounds is appealing to medicinal chemists since it involves a nitro ester-like bond **26b** (Figure 12). All of these reactions had variable yields, ranging from 46% in the case of (R)-ibuprofen and 90% for (S)-ibuprofen. [50].

There is a wide variety of options when it comes to ·NO donors since most are organic nitrocompounds (sometimes mentioned as organic nitrates), but some others are organometallic coordination complexes such as sodium nitroprusside. While this latter compound may be dangerous due to the cyano groups, it is also restricted to use within a hospital because of its potent and fast vasodilatory effect caused by a substitution reaction of the nitrosyl (·NO) by a water molecule (Figure 13) [51]. To overcome this problem, some ruthenium complexes were analyzed for a more controlled realization of ·NO [52], in some cases, under visible light (445 nm), showing that [RuNOPy_2_(NO_2_)_2_OH]·H_2_O (a known ·NO donor) has promising results under this conditions. Unlike nitroprusside **27a**, [RuNOPy_2_(NO_2_)_2_OH]·H_2_O **28a** releases ·NO slowly when exposed to visible light. A slower and more controlled release of ·NO is desired since pharmaceuticals developed from this type of molecule might have fewer side effects and may be used in household treatments [43]. Coordination complexes, often overlooked by medicinal chemists, open a new family of potential biologically active molecules as candidates for drug development. 

## 5. Antitumoral Activity

Among all diseases that humans are subjected to, cancer is considered one of the most severe, as up to today, there are no effective drugs or methods able to treat it entirely. Proliferative diseases are multifactorial and usually complex in terms of treatment. It affects several tissues and it is correlated to many effects at cellular and molecular levels [53,54]. One characteristic that can be used for developing new drugs is the fact that cancer tissue is usually in some degree of hypoxia, so it becomes a target for some nitro aromatic hypoxia-activated prodrugs [55,56]. Heterocycle compounds often show biological activity, particularly the ones with nitrogen and oxygen as the heteroatoms. Benzopyrans, O containing heterocycles, gained attention among researchers due to their high reactivity and accessibility and because these are naturally occurring compounds. A NO_2_ group in the C3 position often increases benzopyrans’ activity, especially their antitumoral effect, making them potential lead compounds for developing new anticancer drugs [57].

Coordination metal complexes with Schiff bases are another class of antitumoral agents. It has been mentioned before that some 4-nitro-1,2-phenylendiamine complexes with the metal Zn(II) show antibacterial activity; however, when the Zn(II) is replaced by Cd(II) the complex (Figure 14) becomes an efficient compound able to inhibit in vitro MCF-7 cell lines, a model for breast cancer [26].

## 6. MAO Inhibitors 

Monoamine oxidase (MAO) is a crucial enzyme linked to the regulation of neurotransmitters such as dopamine, which is related to several diseases of the central nervous system such as Parkinson’s disease, depression and anxiety among others. MAO removes the neurotransmitters norepinephrine, serotonin and dopamine, so inhibition is desired in order to restore dopamine concentration [58]. Hydrazothiazole derivatives with a meta-nitro phenyl substituent **29a–e** exemplify an important pharmacophore for inhibition of MAO which can be a therapeutic target for the treatment of neurodegenerative diseases. A total of thirty-seven compounds containing the afford mentioned pharmacophore were synthesized and properly characterized in order to assess inhibition of human MAO, especially the B isoform (Figure 15) [59]. In vitro experiments were carried out in order to establish the IC_50_ minimum concentration for MAO inhibition; IC_50_ in the range of 0.0018–0.0068 μM for the most potent compounds. Molecular docking results indicated a preference for the inhibition of MAO B over the A isoform in all compounds. These molecules were oriented with the nitro phenyl group towards the FAD cofactor and direct interaction among the NO_2_ moiety and the side chains of Thr201 and Glu84 located at the entry of the active site. The thiazole ring also showed high energy stacking interactions with Tyr326 and Trp119. [59].

## 7. Conclusions 

The nitro group, both as a pharmacophore or a moiety present in the structure, has a wide variety of biological activities as it can interact with many biomolecules inhibiting enzymes, or participating directly in cellular signaling, although the most reported activity corresponds to antimicrobial properties. In most cases, nitrated derivatives of already existing pharmaceuticals are being developed in order to increase activity or decrease side effects, such as the case of nitrated NSAIDs. The NO_2_ is highly electron-withdrawing, so it has a deep effect on the polarity of the corresponding molecules favoring in some cases, interaction with proteins. The electron-withdrawing effect observed in aromatic rings may also play an important role when it comes to developing new synthetic routes, as it favors certain positions for electrophilic substitution reactions. On the other hand, nitro compounds undergo redox reactions within the cell and can release NO as a result. For these reasons, the nitro group is rarely a pharmacophore by itself, and when it does, it is attributed to its toxic effects related to the disruption of oxidative stress mechanisms, as in the case of antiparasitic agents. NO_2_ substituents can also enhance inhibition of target biomolecules such as proteins or enzymes due to its electron-rich environment favoring interaction with some amino acids such as threonine and glutamine. When it comes to metal complexes, it is difficult to assess whether the nitro group is directly responsible for the antitumoral activity or if it is the metal the moiety that causes cytotoxicity. The field of computational chemistry has become a significant tool for a better understanding of interactions between NO_2_ and biological targets such as proteins and enzymes, demonstrating the wide potential of this functional group not only as a pharmacophore/toxicophore but also as an important moiety in the resulting molecule. Despite a vast number of biological effects attributed to the nitro group, new research is still needed for the development of new active and effective compounds. Many of the structures presented in this work are good candidates for future research as they are classified as lead compounds, so their pharmacokinetic properties are not yet known. 

## Figures and Tables

**Figure 1 pharmaceuticals-15-00717-f001:**
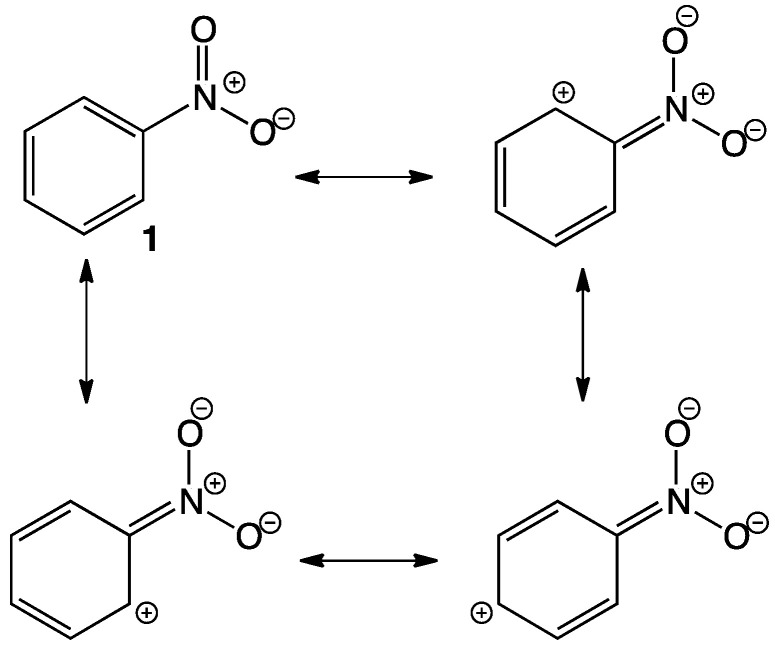
Resonance structures in nitrobenzene.

**Figure 2 pharmaceuticals-15-00717-f002:**
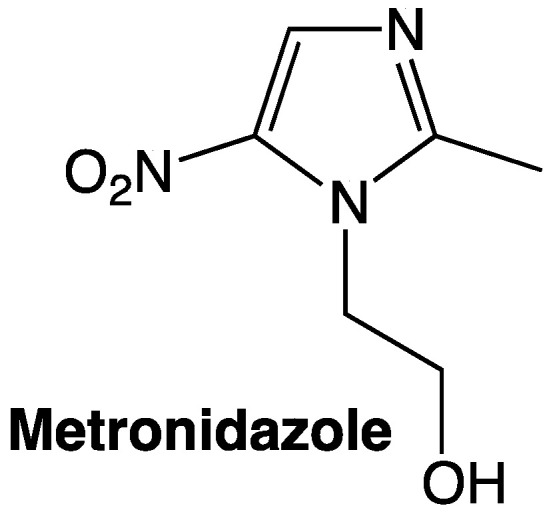
Metronidazole as a 5-nitroimidazole.

**Figure 3 pharmaceuticals-15-00717-f003:**
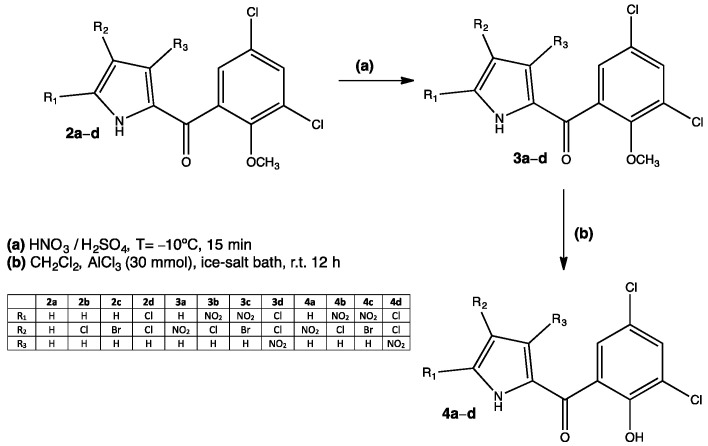
Nitrated pyrrolomycins with potential antibacterial activity [24].

**Figure 4 pharmaceuticals-15-00717-f004:**
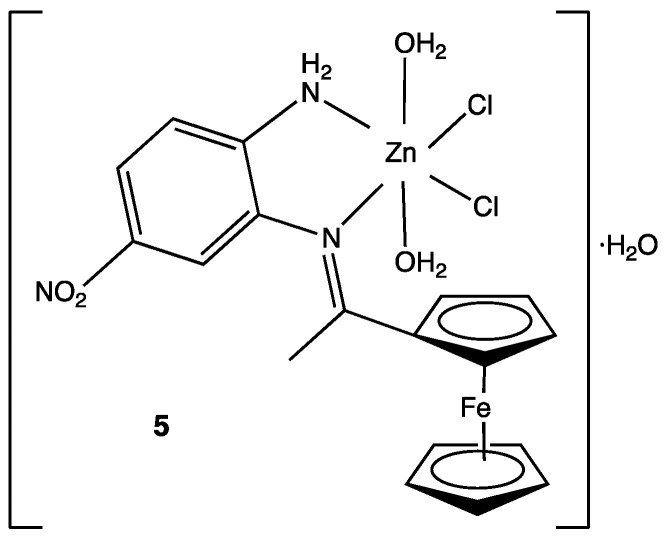
Structure of the Zn(II) complex against *Streptococcus mutans* [26].

**Figure 5 pharmaceuticals-15-00717-f005:**
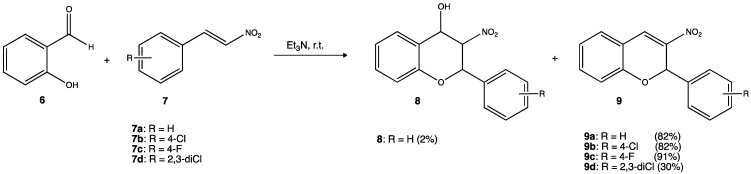
Nitroderivatines containing one F and/or two Cl atoms showed the best activity against *S. aureus* (15.6–62.5 μg/mL) and *Candida* sp. (15–500 62.5 μg/mL), respectively [27].

**Figure 6 pharmaceuticals-15-00717-f006:**
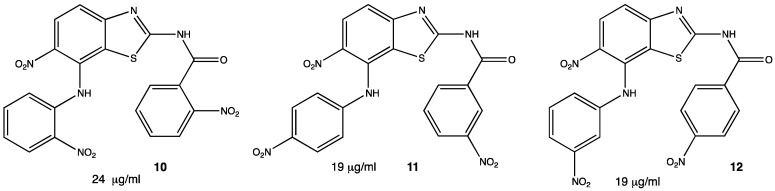
Benzothiazole derivatives against *P. aeruginosa* [28].

**Figure 7 pharmaceuticals-15-00717-f007:**
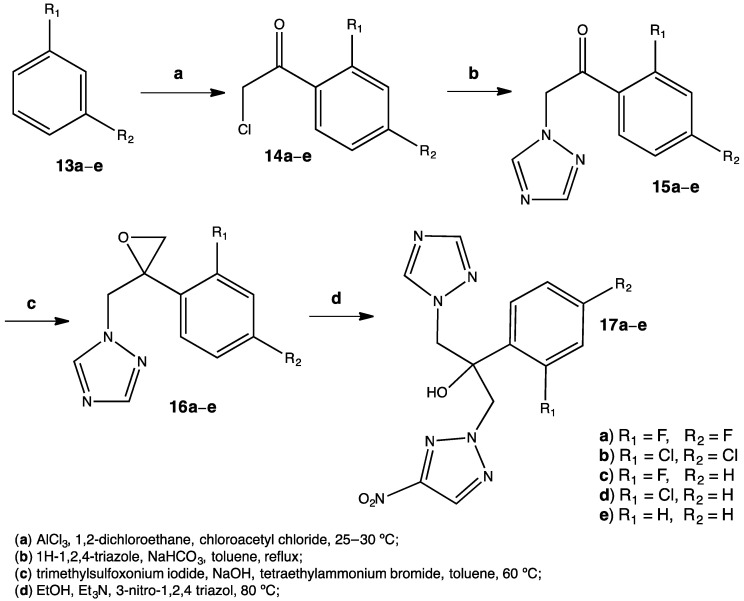
The presence of a nitrotriazole moiety in the fluconazole derivatives enhances antifungal properties [29].

**Figure 8 pharmaceuticals-15-00717-f008:**
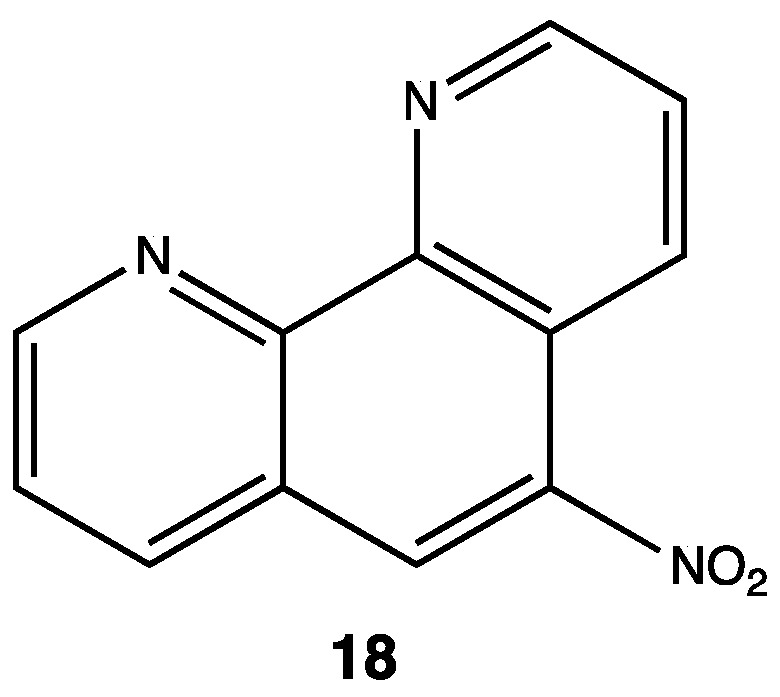
5-nitro-1,10-phenantroline compounds tested against *Mycobacterium tuberculosis* [31].

**Figure 9 pharmaceuticals-15-00717-f009:**
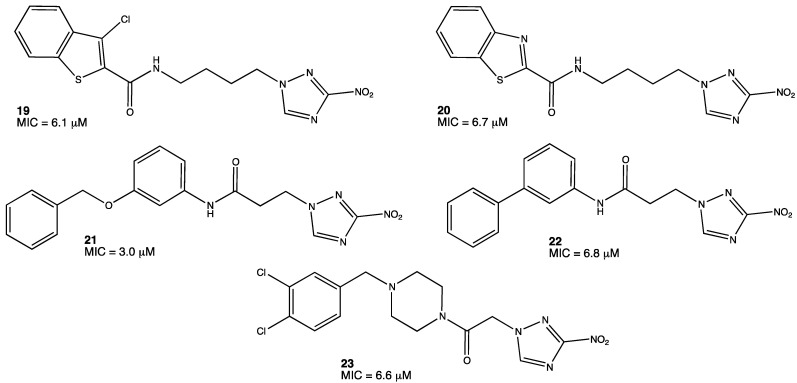
Nitrotriazole derivatives with antitubercular activity [34].

**Figure 10 pharmaceuticals-15-00717-f010:**
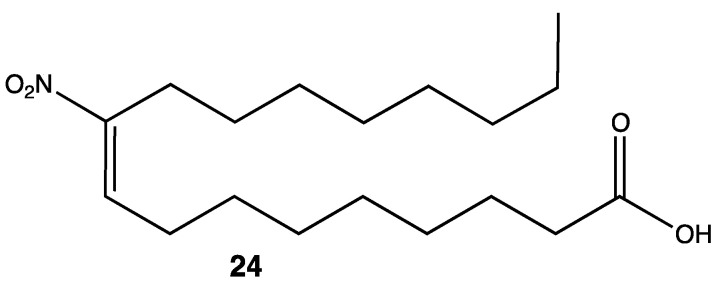
Nitrated fatty acid with anti-inflammatory activity [38].

**Figure 11 pharmaceuticals-15-00717-f011:**
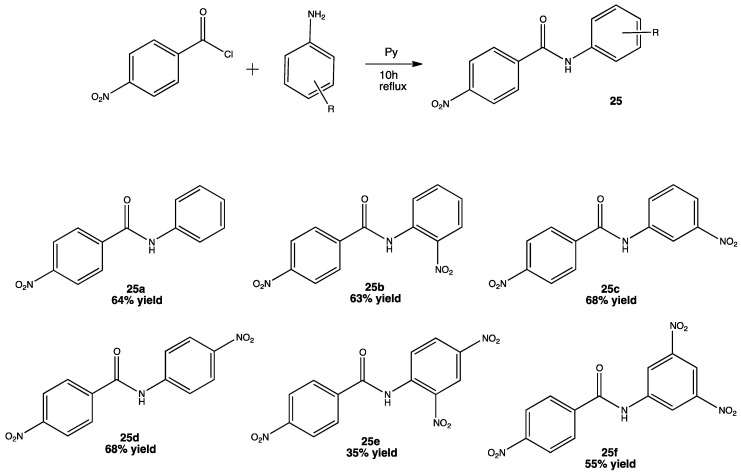
Compound **25f** showed a potent inhibition of pro-inflammatory cytokines making it a good candidate as a lead compound for further research [41].

**Figure 12 pharmaceuticals-15-00717-f012:**
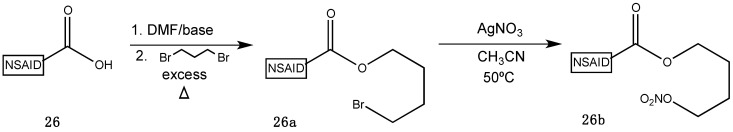
Synthetic pathway for ·NO donor NSAIDs.

**Figure 13 pharmaceuticals-15-00717-f013:**
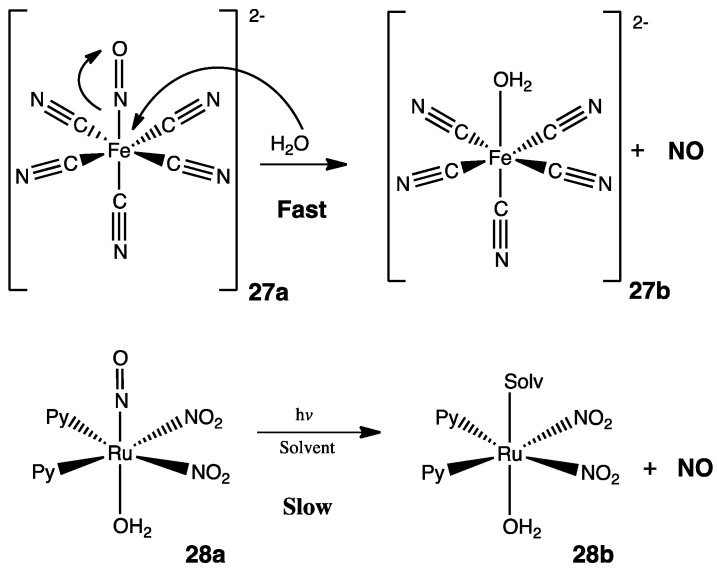
The mechanisms involving the realization of ·NO in coordination complexes [43,51].

**Figure 14 pharmaceuticals-15-00717-f014:**
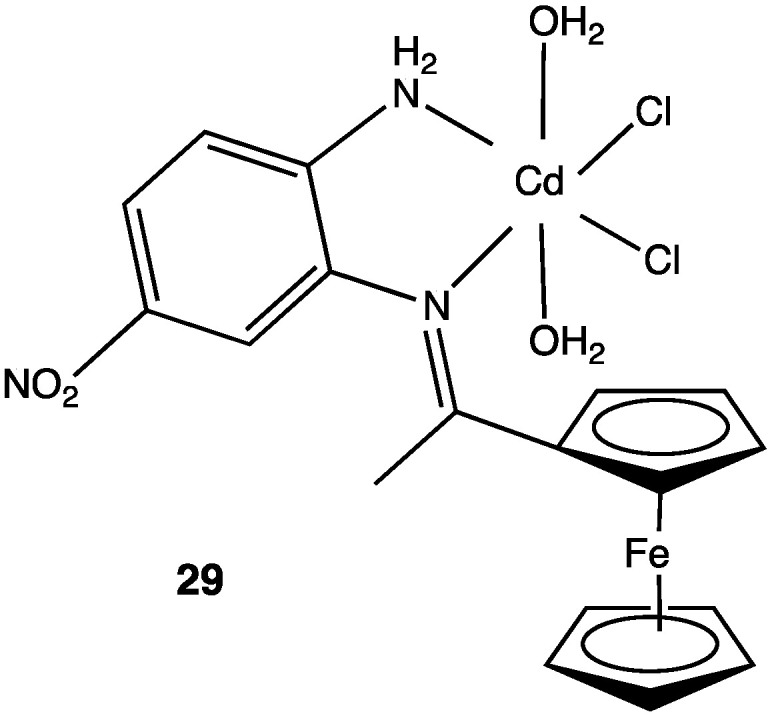
Cd(II) complex with antitumoral activity IC_50_ 50 μg/mL.

**Figure 15 pharmaceuticals-15-00717-f015:**
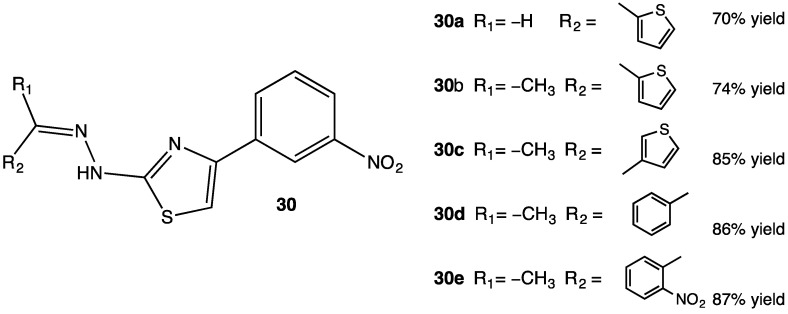
Structures of compounds showing a high and selective inhibition of MAO B [59].

## Data Availability

No new data were created or analyzed in this study. Data sharing is not applicable to this article.

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
