# Peer review of "The Diverse Biological Activity of Recently Synthesized Nitro Compounds"

_pharmaceuticals, 2022, doi:10.3390/ph15060717_

Round 1

Reviewer 1 Report

The manuscript entitles: “The diverse biological activity of recently synthesized nitro compounds”. This is standard review. Authors presented the latest reports on compounds with a nitro moiety with potential application in biology. Undoubtedly an interesting work, however the material needs some little improvements:

1. Authors should correct notation of “figures” in text (capital letter “F”).

2. Authors should correct notation of “Gram-positive” or “Gram-negative” (lines 113, 114, 126, 142).

3. Authors should correct notation of Latin names in italics (lines 149, 202).

4. Authors should correct editorial mistakes, e.g.: typos or double spaces (lines 112, 202, 207), IC50 with subscript (line 317).

5. Authors should correct notation of “compound 6” (line 245). This is the compound of 25f.

6. Authors should correct notation of “NO” of the section Vasodilatory activity. Nitric oxide is a radical. Applicable record is ˙NO.

7. Authors should correct record the "N", "O " in line 294 (italics).

8. I propose to add the valence of metals Ni and Cd (lines 117, 305).

I will be happy to recommend the publication of this paper after this correction. In my opinion, 55 references in the review is too scarce and therefore in the future the authors should, when writing a review, investigate the available literature much more.

Author Response

Response to Reviewer 1 Comments

Point 1:  

Authors should correct notation of “figures” in text (capital letter “F”)

Response 1: All notations have been changed for a capital “F”.

Point 2:

Authors should correct notation of “Gram-positive” or “Gram-negative” (lines 113, 114, 126, 142).

Response 2: All notations have been corrected

Point 3:

Authors should correct notation of Latin names in italics (lines 149, 202).

Response 3: Latin names have been corrected  

Point 4:

Authors should correct editorial mistakes, e.g.: typos or double spaces (lines 112, 202, 207), IC50 with subscript (line 317).

Response 4: Double spaces, several typos and subscripts have been corrected.

Point 5:

Authors should correct notation of “compound 6” (line 245). This is the compound of 25f.

Response 5: The correct compound name has been added

Point 6:

Authors should correct notation of “NO” of the section Vasodilatory activity. Nitric oxide is a radical. Applicable record is ˙NO.

Response 6: All nitric oxide notation has been changed indicating its radical nature

Point 7:

Authors should correct record the "N", "O " in line 294 (italics).

Response 7: The name of the atoms has been changed  

Point 8:

 I propose to add the valence of metals Ni and Cd (lines 117, 305).

Response 8: The valence of these metals has been added.  

Reviewer 2 Report

Manuscript Number: Pharmaceuticals-1728954

entitled: The diverse biological activity of recently synthesized nitro compounds

I had great pleasure reviewing this article. This manuscript is suitable for Pharmaceuticals after considering the below comments:

  1. General: Please add the activity of every compound if in the title of Scheme or Fig. Authors mentioned it. Please add the yields of every product if the authors say about synthetic routes. Only Figure 8. Nitrotriazole derivatives with antitubercular activity [33] presented the activity of reported molecules.
  2. Page 2. “Figure 1. Resonance structures in nitrobenzene” should be modified. In the middle of the Scheme, the structure R-NO2 could be deleted. The substituent „R” is not defined, and „1” belongs only to one resonance structure.
  3. Typos; Abstract “This The” Page 3 “The Bromine.”
  4. “Figure 2.” should be modified. Possible, the most important NO2 substituents should be presented. R as hydrogen could be skipped. „Nitrated pyrrolomycins with enhanced antibacterial activity” possibly activity should be added.
  5. Conclusions „The nitro group” or the compounds with the nitro group? „NO2 substituents” how many different NO2 substituents are authors thinking about?

Author Response

Point 1:

General: Please add the activity of every compound if in the title of Scheme or Fig. Authors mentioned it. Please add the yields of every product if the authors say about synthetic routes. Only Figure 8. Nitrotriazole derivatives with antitubercular activity [33] presented the activity of reported molecules.

Response 1: The activity os nitrocompounds have been added as well as the corresponding reaction yields.

Point 2:

Page 2. “Figure 1. Resonance structures in nitrobenzene” should be modified. In the middle of the Scheme, the structure R-NO2 could be deleted. The substituent „R” is not defined, and „1” belongs only to one resonance structure.

Response 2: Figure 1 has beed modified accordingly

Point 3:

Typos; Abstract “This The” Page 3 “The Bromine.”

Response 3: All typos have been corrected

Point 4:

“Figure 2.” should be modified. Possible, the most important NO2 substituents should be presented. R as hydrogen could be skipped. „Nitrated pyrrolomycins with enhanced antibacterial activity” possibly activity should be added.

Response 4: Figure caption has been modified (an extra Figure was added) in sequence and the word “potential” has been added. We consider that keeping the substituent table as it is has a good impact since it allows the reader to undestand the reactivity of the C1 position. A typo in this section has also been corrected.

Point 5:

Conclusions „The nitro group” or the compounds with the nitro group? „NO2 substituents” how many different NO2 substituents are authors thinking about?

Response 5: Conclusions have been modified in order to establish wether the nitro group is the pharmacophore or a moiety of the molecule. This sections has been better explained.

Reviewer 3 Report

Authors of "The diverse biological activity of recently synthesized nitro compounds" need to improve the manuscript. Review should be done for certain time period and "recently synthesized" does not cover any time period.

Most of presented results should contain also results for reference compounds, here this part was omitted and needs improvments.

There is no classification of presented nitro compounds and most importantly number of derivatives is very low.

I suggest to explore more deeply past publications related to nitro derivatives.

Author Response

Point 1: Authors of "The diverse biological activity of recently synthesized nitro compounds" need to improve the manuscript. Review should be done for certain time period and "recently synthesized" does not cover any time period.

Response 1: The date and type of reports analyzed is now mentioned in the introduction section.

Point 2: Most of presented results should contain also results for reference compounds, here this part was omitted and needs improvments.

Response 2: Most compounds mentioned are being investigated as lead compounds, so there is not a particular molecules to be compared with. In some other nitrocompounds, a comparison has been added regrading already existing pharmaceuticals.

Point 3: There is no classification of presented nitro compounds and most importantly number of derivatives is very low.

Response 3: Classification of nitrated compounds is based on their general activity. The role of the nitro group is now described wether it acts as pharmacophore/toxicophore or as moiety improving biological activity/pharmacokinetics/interactions with target biomolecules.

Point 4: I suggest to explore more deeply past publications related to nitro derivatives.

Response 4: Some references from previous reports (before 2017) have been added, as well as some other reviews with related topics.  

Reviewer 4 Report

The authors have presented a review article on recent developments of new bioaсtive nitro compounds. The manuscript is well-written and structured, and covers substantial amount of relevant papers from 2016-2022. To my opinion, the provided analysis of these findings may be of interest to the readers of Pharmaceuticals Journal despite several very similar works have been published recently (like doi 10.1021/acs.jmedchem.8b00147, 10.3390/ph11020054, 10.3390/molecules25081909). I suggest the authors should cite them to provide the readers with better understanding of the situation in this research field. My recommendation is to accept manuscript after some improvements of the text (conclusion and reference sections; also, several typos should be corrected) and figures - see attached pdf-file, yellow color marks and corresponding comments.

Author Response

Point 1:

The authors have presented a review article on recent developments of new bioaсtive nitro compounds. The manuscript is well-written and structured, and covers substantial amount of relevant papers from 2016-2022. To my opinion, the provided analysis of these findings may be of interest to the readers of Pharmaceuticals Journal despite several very similar works have been published recently (like doi 10.1021/acs.jmedchem.8b00147, 10.3390/ph11020054, 10.3390/molecules25081909). I suggest the authors should cite them to provide the readers with better understanding of the situation in this research field. My recommendation is to accept manuscript after some improvements of the text (conclusion and reference sections; also, several typos should be corrected) and figures - see attached pdf-file, yellow color marks and corresponding comments.

Response 1: All suggestions and corrections marked in yellow have been attended properly.

The conclusions sections has been further discussed mentioning advantages and disadvantages of nitrated compounds as well as the role of computational chemistry in the search for new active compounds. References mentioned were already present in the manuscript, and the third one has been added.

Round 2

Reviewer 2 Report

Dear Authors,

The authors conducted essential studies. This is an interesting article. The current version of the manuscript is very well developed and well written. The data presented are new and relevant. The current version is much better, so the manuscript is suitable for publication in its current form.

Reviewer 3 Report

Manuscript has been improved and suitable corrections were made according to reviewer suggestions.

Reviewer 4 Report

The authors have adressed all my comments. I recommend to publish this manuscript in its present form.